# The Influence of Ligands on the Pd-Catalyzed Diarylation of Vinyl Esters

**DOI:** 10.3390/molecules29102268

**Published:** 2024-05-11

**Authors:** Anna Brodzka, Dominik Koszelewski, Anna Trzeciak, Lena Ruzik, Malgorzata Grela, Ryszard Ostaszewski

**Affiliations:** 1Institute of Organic Chemistry, Polish Academy of Sciences, Kasprzaka 44/52, 01-224 Warsaw, Polandmalgorzta.grela@icho.edu.pl (M.G.); 2Faculty of Chemistry, University of Wrocław, F. Joliot-Curie 14, 50-383 Wrocław, Poland; anna.trzeciak@uwr.edu.pl; 3Faculty of Chemistry, Warsaw University of Technology, Noakowskiego 3, 00-664 Warsaw, Poland; lena.ruzik@pw.edu.pl

**Keywords:** diarylation, palladium, water medium

## Abstract

The impact of ligands on the palladium-catalyzed 1,2-diarylation reaction course is presented. The application of Pd-dmpzc as a catalyst provides an efficient, chemoselective and sustainable protocol for the synthesis of valuable 1,2-diphenylethyl acetates. The reaction is conducted in water under mild conditions. Reaction products can be easily separated from the reaction mixture and catalyst by simple extraction. What is more, the rational choice of catalyst significantly reduces the leaching of the metal into the product and its contamination (0.1 ppm). Efficient phase separation and ultralow Pd leaching enable the reuse of the water medium containing the Pd-dmpzc catalyst several times without a significant loss of activity and with even higher selectivity (from 95% to 100% in the third cycle). The recyclability of both the catalyst and the reaction medium together with high chemoselectivity and low palladium leaching reduces the amount of waste and the cost of the process, exhibiting an example of a sustainable and green methodology.

## 1. Introduction

Carbon–carbon cross-coupling reactions are one of the most powerful synthetic tools from the points of view of medicinal chemistry and drug discovery. Most of these transformations are mediated by transition metal catalysts—in particular, palladium-based species—as they allow us to obtain a diverse range of products [1]. However, palladium has an undesired tendency to remain in the final products. Special attention has been paid to the purity of compounds required by pharmacy and medicinal chemistry. A very low limit of heavy-metal (cations) contaminations, which is now below 10 ppm and 1 ppm, for oral and parenteral drug substances, respectively, is excluded from most synthetic methods [2]. Such low levels of palladium permitted in pharmaceutically relevant compounds is due to this metal’s ability to coordinate with proteins, DNA and other macromolecules inhibiting various cellular functions [3]. Therefore, one of the most challenging problems in modern organic chemistry and technology is the design of synthetic methods free from metal impurities.

Our ongoing interest in the design of environmentally benign protocols for the synthesis of biologically relevant compounds [4,5] led us to the development of a palladium-catalyzed 1,2-diarylation of vinyl esters in an aqueous medium [6]. This protocol utilizes arylboronic acids and inexpensive palladium acetate as a catalyst and offers ready-made one-step access to 1,2-diarylethyl esters. These products and their derivatives are an important class of compounds for medicinal chemistry (Figure 1). This includes antineoplastic combretastatin [7] and other molecules with antiviral (against Rhinovirus (RV) [8] and SARS-CoV-2 [9]), antitubercular [10] or fungicidal [11] properties. However, the previously developed protocol does not fulfil the requirements of the limits of palladium contamination. Thus, to overcome these limitations, we decided to study the impact of a ligand structure on the leaching of palladium into the products and the reaction course. The rational design of the catalyst, where the ligand prevents the metal from leaching into products, can also enable the reuse of the catalyst.

## 2. Results and Discussion

To study the efficiency of various palladium catalysts and palladium leaching in products, we chose a previously developed diarylation reaction [6] (Figure 1). As model substrates, we chose phenylboronic acid (**1**) and vinyl acetate (**2**). The reaction was conducted in water at room temperature in the presence of 1,4-benzoquinone (BQ) as a re-oxidant and a palladium catalyst over 24 h. At first, we tested the impact of various palladium species on the reaction course. We tested 15 different complexes of palladium (Figure 2) [12,13,14,15,16,17,18,19] and previously used palladium (II) acetate.

Based on the previously obtained data [6], the studied reaction led to the formation of 1,2-diphenylethyl acetate (**3**) as a main product and two by-products—stilbene (**4**) and 2-phenyl-1,4-benzoquinone (**5**). The majority of the re-oxidant—benzoquinone—was reduced to hydroquinone, which remains in a water phase after the extraction of reaction products by ethyl acetate. Herein we investigated the chemoselectivity of the studied catalysts as well as their efficiency. The results are presented in Table 1.

To develop the sustainable process, the application of environmentally friendly reaction conditions in not sufficient. Thus, in the next step of the studies, we focused our attention on the impact of the palladium catalyst structure on the contamination of the product by metal. As 1,2-diphenylethyl acetate and its derivatives are pharmaceutically relevant compounds [7,8,9,10,11], it is important to develop a synthetic protocol that allows us to obtain little-contaminated products using palladium species. For the most efficient catalysts, the leaching of palladium in product **3** was analyzed via ICP-MS/MS (Table 2).

The obtained results show that the type of ligand used has a crucial impact on the palladium leaching in the final product (**3**) (Table 2). The highest leaching of palladium (26 ppm) was observed for previously used palladium (II) acetate (Table 2, entry 1) and structurally similar palladium (II) stearate (**PP015**) (17 ppm, Table 1, entry 6). The stability of these complexes is lower; thus, the leaching of palladium into the solution is observed. Also, the application of bis(acetonitrile)palladium dichloride (**PP10**) does not fulfill the pharmacopeia requirements (12 ppm, Table 1, entry 4). The change of ligand significantly reduces the palladium leaching in the product. The best results (below 5 ppm) were obtained for catalysts **PP05**, **PP08** and **PP13** (Table 1, entries 2, 3, 5). However, the ultralow levels of metal impurities (below 0.5 ppm) were measured for **PP08** and **PP13** (Table 1, entries 3, 5). What is more, catalyst **PP13** is not only the most chemoselective catalyst (Table 1, entry 14) but also provides products with an ultralow level of palladium content. The intermolecular hydrogen bonds, responsible for the formation of the network structure of catalyst **PP13**, can have a stabilizing effect. This may favor more persistent binding of palladium, which limits its leaching. The stability of the complex (which limits leaching) also sustains catalytic activity and results in higher yields. The reduction of metal impurities as well as side products simplifies the purification procedures and significantly reduces the amount of waste and costs.

As the Pd-dmpzc catalyst (**PP13**) revealed the lowest metal leaching into the product, we also investigated the possibility of reusing the **PP13** catalyst (Figure 3). The studied 1,2-diarylation reaction was conducted in water, and the products were separated from the reaction mixture by simple extraction with ethyl acetate. It is important to note that the catalyst **PP13** remains in the water phase, which enables the reuse of the medium containing the catalyst. The studied catalytic system was reused in five cycles for the diarylation reaction of vinyl acetate with phenylboronic acid. After each catalytic cycle, the products were separated via extraction with organic solvent, and the fresh portion of substrates was added to an aqueous phase containing a catalyst. It is also worth noting that the amount of by-product **4** decreased with each cycle—in the second run, only 2% of stilbene was formed, and it disappeared completely in subsequent cycles. The recyclability studies reveled a slight decrease in product **3**’s yield, which could have been caused by the remaining reagents in the water phase (such as hydroquinone, boric acid or unreacted substrates). It is worth noting that the selectivity of the reaction increases in subsequent cycles—up to 100% in the third run (Figure 3).

Finally, to prove the generality of the presented catalyst, some additional experiments were performed (Figure 2). When 3,4-dimethoxyphenylboronic acid and vinyl acetate were used as substrates, diarylation product **6** was obtained with 52% yield (33% for previously used palladium acetate). Then, we changed the vinyl ester from acetate to benzoate and obtained product **7,** with nearly a twofold higher yield than that for Pd(OAc)_2_. These results indicate that catalyst **PP13** is not only chemoselective, highly active and recyclable but also works well for various substrates.

## 3. Materials and Methods

### 3.1. General Experimental Details

Unless otherwise stated, all reagents were purchased from commercial sources (incl. Sigma-Aldrich Poznan, Poland (Merck) and Tokyo Chemical Industry, Tokyo, Japan) and were used without additional purification. The hexane mixture and water were distilled prior to use. Other solvents (analytical grade) were applied without extra purification and drying. All reactions were performed in dry glass flasks or vials. Solvents were evaporated under reduced pressure utilizing a rotary evaporator equipped with a water bath. Reactions were checked and preliminarily analyzed by TLC on Merck silica gel plates (60 Å, F254). To visualize compounds on the plates, a UV lamp (λ = 254 nm) and TLC stains (potassium permanganate or cerium(IV) sulfate) were applied.

All compounds were purified by flash or column chromatography. Flash chromatography was conducted on a CombiFlash^®^ Rf 200. Elution was carried out in a hexane-ethyl acetate system. NMR spectra were recorded in CDCl_3_ on a Bruker spectrometer at 400 MHz. The chemical shifts were reported in ppm relative to the deuterated solvent or TMS standard signal, and the coupling constants (J) were given in Hz. For the quantification of palladium, an Agilent 8900 ICP Triple Quadrupole Mass Spectrometer (Tokyo, Japan), as an element-specific detector, was used. The spectrometer was equipped with Pt-cones in the interface. The position of the torch and the nebulizer gas flow were adjusted daily, with emphasis placed on the decrease in the level of CeO^+^ (below 0.2%), to minimize the risk the polyatomic interferences caused by oxides. The RF power was 1540 W, the nebulizer gas flow was 1.01 L min^−1^, and the collision/reaction gas flow (hydrogenium in ICP-MS/MS) was 5.0 mL min^−1^. The total concentrations of selected metals were calculated as a result of monitoring the mass/charge ratios: ^95^Pd, registered in the on-mass mode after the production in plasma of singly positively charged ions. The working conditions were optimized daily using a 10 µg L^−1^ solution of ^7^Li^+^, ^89^Y^+^ and ^209^Bi^+^ in 2% (*v*/*v*) HNO_3_. Microwave digestion for the mineralization of samples was conducted using a Speedwave^®^ four Berghof, (Germany). 

### 3.2. General Procedure for α,β-Homodiarylation of Vinyl Esters

A glass snap-cap vial (10 mL), provided with a stir bar, was loaded with solid reagents: arylboronic acid (**1**; 3 mmol), palladium catalyst (**PP01**–**PP15**) (5 mol%), 1,4-benzoquinone (0.9 mmol), 4 mL of water, and vinyl ester (**2**, 0.75 mmol). The vial was closed, and the reaction mixture was stirred at room temperature for 24 h. Next, ethyl acetate (10 mL) was added, resulting in a two-phase solution. The organic phase was separated, and the remaining phase was extracted with 20 mL ethyl acetate. The combined organic layers were dried with MgSO_4_, and residuals of the solvent were removed under reduced pressure. The crude product was purified by column chromatography on silica gel using hexane/AcOEt as an eluent. The yields of the derivatives are shown in Table 1. The structures of the products were identified by their ^1^H and ^13^C NMR in electronic support information (ESI), and known compounds were compared with the literature data.

1,2-diphenylethyl acetate (**3**). The product was isolated as a yellow oil from a silica column eluted by EtOAc/hexanes, R_f_ = 0.62 (EtOAc/hexanes 20:80); ^1^H NMR (400 MHz, CDCl_3_): *δ* 7.44–7.24 (m, 8H), 7.23–7.13 (m, 2H), 6.05 (dd, *J* = 7.9, 6.0 Hz, 1H), 3.28 (dd, *J* = 13.7, 7.9 Hz, 1H), 3.14 (dd, *J* = 13.7, 6.1 Hz, 1H), 2.06 (s, 3H).; ^13^C{H} NMR (100 MHz, CDCl_3_): δ 170.1, 140.2, 137.1, 129.6, 128.4, 128.3, 128.0, 126.7, 126.6, 76.7, 43.0, 21.1. The ^1^H and ^13^C NMR data were in accordance with those reported in the literature [6].

Stilbene (**4**). The product was isolated as a white solid (m.p. 123–124 °C) from a silica column eluted by EtOAc/hexanes (10:90); ^1^H NMR (400 MHz, CDCl_3_): δ 7.56–7.47 (m, 4H), 7.40–7.34 (m, 4H), 7.28–7.23 (m, 2H), 7.12 (s, 2H); ^13^C{^1^H} NMR (100 MHz, CDCl_3_) δ 137.4, 128.7, 128.6, 127.6, 126.5. The ^1^H and ^13^C NMR data were in accordance with those reported in the literature [6].

2-phenyl-1,4-benzoquinone (**5**). The product was isolated as a brown solid from a silica column eluted by EtOAc/hexanes (30:70); ^1^H NMR (400 MHz, CDCl_3_): δ 7.49–7.42 (m, 5H), 6.87–6.80 (m, 3H); ^13^C{H} NMR (100 MHz, CDCl_3_): δ 187.6, 186.6, 145.9, 137.0, 136.2, 132.2, 130.1, 129.2, 128.5. The ^1^H and ^13^C NMR data were in accordance with those reported in the literature [20].

1,2-bis(3,4-dimethoxyphenyl)ethyl acetate (**6**). The product was isolated as a brown solid from a silica column eluted by EtOAc/hexanes, R_f_ = 0.35 (EtOAc/hexanes 40:60); ^1^H NMR (400 MHz, CDCl_3_): δ 6.78–6.64 (m, 4H), 6.65 (dd, J = 8.1, 2.0 Hz, 1H), 6.50 (d, J = 2.0 Hz, 1H), 5.78 (t, J = 7.0 Hz, 1H), 3.79 (s, 3H), 3.77 (s, 3H), 3.77 (s, 3H), 3.71 (s, 3H), 3.06 (dd, J = 13.7, 7.3 Hz, 1H), 2.91 (dd, J = 13.7, 6.6 Hz, 1H), 1.96 (s, 3H); ^13^C{H} NMR (100 MHz, CDCl_3_): δ 170.1, 148.8, 148.6, 147.7, 132.5, 129.5, 121.7, 119.3, 112.8, 111.0, 110.9, 110.2, 76.7, 55.9, 55.8, 55.7, 42.4, 21.3. The ^1^H and ^13^C NMR data were in accordance with those reported in the literature [6].

1,2-diphenylethyl benzoate (**7**). The product was isolated as a white solid from a silica column eluted by EtOAc/hexanes, R_f_ = 0.65 (EtOAc/hexanes 10:90); ^1^H NMR (400 MHz, CDCl_3_): *δ* 8.03–7.92 (m, 2H), 7.49–7.47 (m, 1H), 7.46–7.34 (m, 2H), 7.30–7.19 (m, 5H), 7.17–7.02 (m, 5H), 6.11 (dd, *J* = 7.6, 6.0 Hz, 1H), 3.28 (dd, *J* = 13.8, 7.6 Hz, 1H), 3.12 (dd, *J* = 13.8, 6.0 Hz, 1H); ^13^C{H} NMR (100 MHz, CDCl_3_): δ 165.6, 140.1, 136.9, 132.9, 130.4, 129.6, 128.4, 128.3, 128.2, 127.9, 126.6, 126.5, 77.3, 43.2. The ^1^H and ^13^C NMR data were in accordance with those reported in the literature [6].

### 3.3. General Procedure for the Synthesis Catalysts ***PP01***–***PP15***

Palladium catalysts **PP01**, **PP03**, and **PP04** were synthesized according to a procedure previously reported by Trzeciak et al. [12]. Palladium catalysts **PP02**, **PP05**, **PP06**, **PP08**, and **PP09** were synthesized according to a procedure previously reported by Trzeciak et al. [13]. Palladium catalyst **PP07** was obtained according to [12], using a dipiperidylmethane ligand synthesized according to a procedure previously reported by Kocięcka et al. [14]. Palladium catalyst **PP10** was synthesized according to a procedure previously reported by Mathews et al. [15]. Palladium catalysts **PP11** and **PP12** were synthesized according to a procedure previously reported by Trzeciak et al. [16]. Palladium catalyst **PP13** was synthesized according to a procedure previously reported by Trzeciak et al. [17,18]. Palladium catalysts **PP14** and **PP15** were synthesized according to a procedure previously reported by Zakrzewska et al. [19].

### 3.4. Reusability and Recovery of the Catalyst

After the first run of the reaction was completed, the product was directly extracted into the organic layer (ethyl acetate), and the catalyst that remained in the aqueous layer was reused for the next cycle of the reaction, following the same procedures as mentioned above.

A glass snap-cap vial (10 mL), provided with a stir bar, was loaded with solid reagents: arylboronic acid (**1**; 3 mmol), 1,4-benzoquinone (0.9 mmol), vinyl ester (**2**, 0.75 mmol) and aqueous layer containing catalyst from previous cycle. The vial was closed, and the reaction mixture was stirred at room temperature for 24 h. Next, ethyl acetate (10 mL) was added, resulting in a two-phase solution. The organic phase was separated, and the remaining phase was extracted with 20 mL ethyl acetate. The combined organic layers were dried with MgSO_4_, and residuals of the solvent were removed under reduced pressure. The crude product was purified by column chromatography on silica gel using hexane/AcOEt as an eluent.

### 3.5. Determination of Palladium Leaching into Product ***3***

To determine the total amount of elements, the sample was digested by microwave-assisted mineralization with a mixture of 2 mL of HNO_3_. The digests, to a final volume of 5 mL with Milli-Q water, were diluted. Further dilutions toward ICP-MS/MS analysis were prepared using 2% (*v*/*v*) nitric acid solution and 10 ng mL^−1^ of yttrium (^89^Y) as an internal standard.

The total concentration of elements in the sample was measured by ICP-MS/MS. The results represent the average amount established for three samples (each measured three times) and show the total concentration of elements. The method’s precision was evaluated by analyzing the ten independent experiment preparations for each metal—the test samples against the internal standard and the %RSD of metals calculated. The accuracy of the obtained data was high and repeatable (%RSD) and was in the range of 1.71–5.12%.

A linearity test was performed to check the capacity of the entire analytical system to display a linear response, and the proportionality of the signal intensity to the relevant concentration of the analyte varied within a certain range. The obtained results’ dependence on the analyte concentration was linear, in the range 0.5 µg L^−1^–100.0 µg L^−1^ for Pd, with an r^2^ above 0.9996.

## 4. Conclusions

In conclusion, we have presented studies on the impact of the structure of the palladium ligand on the chemoselectivity of vinyl ester diarylation reactions and metal leaching into products. The reaction was conducted at room temperature in water under mild conditions, which makes it attractive from environmental point of view. The application of Pd-dmpzc (**PP013**) as a catalyst not only provides the best chemoselectivity (elimination of by-products) but also reduces the leaching of palladium into the product to an ultralow level (0.1 ppm); this considerably diminishes the amount of generated waste and the cost of the whole procedure. It is also worth noting that the change of ligand from the previously used acetate to dmpzc resulted in great decrease in metal contamination in final product. This, and the water solubility of complex **PP13**, allow us to develop the methodology based on the recyclability of the catalyst and water medium several times without a significant loss in yield. The same reaction medium containing the catalyst was used up to five times, with excellent chemoselectivity. Reaction products were easily separated from the reaction mixture containing the catalyst and other reagents by a simple extraction with ethyl acetate, which also simplifies the purification procedure. It is important to note that the selectivity of the reaction catalyzed by **PP13** increases in subsequent cycles, reaching 100% in the third cycle. To the best of our knowledge, such an improvement in selectivity in the next cycles was not observed. The use of an aqueous medium and the recyclability of the catalyst and medium at least five times reduces the amount of generated waste and the cost of the process, exhibiting an example of a sustainable and green methodology.

## Data Availability

Data are contained within the article and Appendix A.

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
