# Peer review of "The Influence of Ligands on the Pd-Catalyzed Diarylation of Vinyl Esters"

_molecules, 2024, doi:10.3390/molecules29102268_

Round 1

Reviewer 1 Report

Comments and Suggestions for Authors

This Article presents the impact of ligand on the palladium-catalyzed oxidative 1,2-diarylation of vinyl acetates with aryl boronic acids. BQ was used as the external oxidant. A bunch of ligated palladium salts were tested under aqueous conditions. It was found that the catalytic system with PP13 (Pd-dmpzc) provided the desired product 1,2-diphenylethyl acetates with excellent selectivity. Moreover, efficient phase separation afforded the product and ultralow Pd leaching was observed (0.1 ppm). In addition, because of the stability, the Pd-dmpzc catalyst in water medium can be reused for several times without significant loss of activity. It can even afford higher selectivity (from 95% to 100% in third cycle). The recyclability of both the catalyst and the reaction medium, the high chemoselectivity, and the low palladium leaching together demonstrated an example of sustainable and green methodology.

However, this work is an extension of the authors’ EJOC paper (Eur. J. Org. Chem. 2021, 6028). Most importantly, this reviewer did not find the interests of this method in synthetic applications because of the narrowness of this method. Therefore, I think this manuscript does not meet the criterion to publish on Molecules.

Specific comments:

1)     The title of this Article is “The influence of ligands on the Pd-catalyzed C-C coupling reactions course”, which looks like a review paper topic. The title should be more specific and accurate.

2)     This palladium-catalyzed diarylation of vinyl esters is an extension of the authors’ EJOC paper. It is pleased to learn that this manuscript dealing with ultralow Pd leaching and excellent results were obtained. This reviewer wonders that what would be the results if the obtained catalyst Pd-dmpzc was used in a specific synthetic route, for example, in the synthesis of Combrestatin.

3)     A major drawback of this method is the incorporation of two same aryl group from the same aryl boronic acids. Is it possible to use different aryl boronic acids, and obtain the product with different aryl groups?

4)     The authors mentioned the high recycle ability of the Pd-dmpzc. It is suggested that the authors provide reasonable explanation of this observation.

5)     In figure 2 the name of pp12, Pd(OAc)2, and other mistakes should be corrected.

Author Response

Firstly, we would like to express our gratitude to Reviewers for their suggestions that allowed us to considerably improve our manuscript.

  • The title of this Article is “The influence of ligands on the Pd-catalyzed C-C coupling reactions course”, which looks like a review paper topic. The title should be more specific and accurate.

Answer: The title was changed to “The influence of ligands on the Pd-catalyzed diarylation of vinyl esters.”

  • This palladium-catalyzed diarylation of vinyl esters is an extension of the authors’ EJOC paper. It is pleased to learn that this manuscript dealing with ultralow Pd leaching and excellent results were obtained. This reviewer wonders that what would be the results if the obtained catalyst Pd-dmpzc was used in a specific synthetic route, for example, in the synthesis of Combrestatin.

Answer: The previously developed palladium-catalyzed diarylation of vinyl esters method published in EJOC has limited synthetic application as products are contaminated by palladium acetate. Now we have focused out attention to develop methodology which can be applied in industry, free from metal impurities. In present work, we show the general synthetic route towards 1,2-diarylethanols, which can be also applied for the synthesis of Combrestatin.

3)     A major drawback of this method is the incorporation of two same aryl group from the same aryl boronic acids. Is it possible to use different aryl boronic acids, and obtain the product with different aryl groups?

Answer: The proposed methodology can be used for the synthesis of compounds with two different aryl groups, however in such case the mixture of 4 various products is formed.

4)     The authors mentioned the high recycle ability of the Pd-dmpzc. It is suggested that the authors provide reasonable explanation of this observation.

Answer: The catalyst Pd-dmpzc catalyst (PP13) remains in water phase after reaction and practically does not leach into the product, provides ultralow level of palladium content. Thus, the amount of catalyst remains on the same level in each cycle. The intermolecular hydrogen bonds, responsible for the formation of the network structure of catalyst PP13, can have a stabilizing effect. This may favour more persistent binding of palladium, which limits its leaching. The stability of the complex (which limits leaching) also sustains catalytic activity and results in higher yields.

5)     In figure 2 the name of pp12, Pd(OAc)2, and other mistakes should be corrected.

Answer: The found errors were corrected.

Reviewer 2 Report

Comments and Suggestions for Authors

In this manuscript, Brodzka and co-workers reported on the impact of ligands on the palladium-catalyzed diarylation of vinyl esters. The catalyst Pd-dmpzc (PP013) not only provides excellent chemoselectivity but also significantly reduces the leaching of palladium into the product to an ultralow level. However, major revisions should be made.

1.     The title should be corrected from 'The influence of ligands on the Pd-catalyzed C-C coupling reactions course.' to 'The influence of ligands on the Pd-catalyzed diarylation of vinyl esters'.

2.     In this reaction, the authors used excessive amounts of phenylboric acid; therefore, there is no need to discuss the formation of the by-product 2-phenyl-1,4-benzoquinone (5).

3.     Authors should provide the structures of PP011-PP013, which would be preferable.

4.     The substrate scope is limited; therefore, the authors should provide more examples to effectively demonstrate the advantages of this reaction strategy.

Author Response

Firstly, we would like to express our gratitude to Reviewers for their suggestions that allowed us to considerably improve our manuscript.

  1. The title should be corrected from 'The influence of ligands on the Pd-catalyzed C-C coupling reactions course.' to 'The influence of ligands on the Pd-catalyzed diarylation of vinyl esters'.

Answer: We are grateful for this opinion. The title was change according to the suggestion.

  1. In this reaction, the authors used excessive amounts of phenylboric acid; therefore, there is no need to discuss the formation of the by-product 2-phenyl-1,4-benzoquinone (5).

Answer: We agree with this opinion. However, in the present work we would like to discuss all the experimental data to show the impact of catalyst used on the reaction course and selectivity.

  1. Authors should provide the structures of PP011-PP013, which would be preferable.

Answer: The structures were added.

  1. The substrate scope is limited; therefore, the authors should provide more examples to effectively demonstrate the advantages of this reaction strategy.

Answer: In present work we focused our attention on the impact of ligand on the reaction course and palladium leaching into products. The aim of our work was to show the general concept  rather than the substrate scope and limitation.

Reviewer 3 Report

Comments and Suggestions for Authors

This manuscript reports on the continuation of the authors previous work (Eur. J. Org. Chem. 2021, 6028-6036) concerning the Pd-catalysed diarylation of vinyl esters by boronic acids in water. They show that by the proper choice of the catalyst, Pd leaching can be reduced considerably and the catalytic system can be reused. It is also shown that its applicability is not restricted to the model reaction, but it can also be used in the reaction of  another boronic acid or vinyl ester derivative. It is also shown that the products can be obtained with  even better yields than with the previously used Pd (OAc)2 catalyst. The methodology gives a real advanvement compared to the provious one so the manuscript deserves publication after some minor corrections.

1) I think the title could be more specific (e.g. The influence of ligands on the Pd-catalyzed diarylation of vinyl esters)

2) The ligand ’pymo’ should be specified (Figure 2)

3 ) Numbers of values of yields should be added to Figure 3

4) There seems to be no supplementary material, so the paragraph ’ Supplementary Materials: The following supporting information can be downloaded at: www.mdpi.com/xxx/s1.’ should be deleted

Some further suggestions for minor points to be corrected.

line 24: The carbon-carbon cross coupling reactions are one of

line 29: Very low limit of heavy metal (cations) contaminations

line 32: this metal’s ability

line 37: led us to the development

line 71: majority of the re-oxidant – benzoquinone

line 81 ..reaction conditions is not sufficient. Thus ……we have focussed our attention

line 100 the catalyst PP13 is not only the most

line 112 the reuse of the medium containing the catalyst

line 136 some additional experiments were

line 138 product 6 was obtained with 52% yield

line 141: not only chemoselective, highly active and recyclable

line 154: rotary evaporator

line 164: The sentence ’ The spectrometer, with the Pt-cones in the interface, was equipped.’ seems to be unfinished

Comments on the Quality of English Language

Minor editing of English language required

Author Response

Reviewer 3: 

Firstly, we would like to express our gratitude to Reviewers for their suggestions that allowed us to considerably improve our manuscript. The found errors were corrected. The revision of manuscript was made.

  1. I think the title could be more specific (e.g. The influence of ligands on the Pd-catalyzed diarylation of vinyl esters)

Answer: We are grateful for this opinion. The title was change according to the suggestion.

  1. The ligand ’pymo’ should be specified (Figure 2)

Answer: The structure of a catalyst was added to Figure 2..

  1. Numbers of values of yields should be added to Figure 3.

Answer: The requested values were added.

  1. There seems to be no supplementary material, so the paragraph ’ Supplementary Materials: The following supporting information can be downloaded at: www.mdpi.com/xxx/s1.’ should be deleted

Answer: The Supplementary Materials were included.

Reviewer 4 Report

Comments and Suggestions for Authors

Article “The influence of ligands on the Pd-catalyzed C-C coupling reactions course” by Brodzka et al. shows impact of ligand on the palladium-catalyzed 1,2-diarylation reaction. The work is well written and done. The leaking of Pd to the product was also studied. Authors fully characterize prepared products of coupling reaction, but do not provide characterization of prepared catalysts. For Figure 2, it would be nice if at least PP13 can be provided as structural formula. For Figure 3, it would be nice if authors can also provide yields of reactions as numbers above the column, as they shown for selectivities. For scheme 2, there are missing closing parentheses for alternative catalysts. It would be nice if section 3.4. is provided with more details. First author in reference 18 is misspelled.

Generally, I would recommend the manuscript for publication after revision.

Author Response

Firstly, we would like to express our gratitude to Reviewers for their suggestions that allowed us to considerably improve our manuscript.

  1. For Figure 2, it would be nice if at least PP13 can be provided as structural formula.

Answer: The structures of catalysts PP11-PP13 were added.

  1. For Figure 3, it would be nice if authors can also provide yields of reactions as numbers above the column, as they shown for selectivities.

Answer: The yields were added.

  1. For scheme 2, there are missing closing parentheses for alternative catalysts

Answer: The closing parentheses were added.

  1. It would be nice if section 3.4. is provided with more details.

Answer: More details were added to the section 3.4.: “After the first run of the reaction was completed, the product was directly extracted into the organic layer (ethyl acetate), and the catalyst which remained in the aqueous layer was reused for the next cycle of the reaction, following the same procedures as mentioned above.

Glass snap cap vial (10 mL), provided with a stir bar, was loaded with solid reagents: arylboronic acid (1; 3 mmol), 1,4-benzoquinone (0.9 mmol), vinyl ester (2, 0.75 mmol) and aqueous layer containing catalyst from previous cycle. The vial was closed and the reaction mixture was stirred at room temperature for 24 hours. Next, ethyl acetate (10 mL) was added, resulting in a two-phase solution. The organic phase was separated, and the remaining phase was extracted with 20 mL ethyl acetate. The combined organic layers were dried with MgSO4 and residuals of solvent were removed under reduced pressure. The crude product was purified by column chromatography on silica gel using hexane/AcOEt as an eluent.”

  1. First author in reference 18 is misspelled.

Answer: The error was corrected.

  1. The ligand ’pymo’ should be specified (Figure 2)

Answer: The structure of catalyst was added.

Round 2

Reviewer 1 Report

Comments and Suggestions for Authors

The authors have revised the manuscript according to the comments raised by the all the reviewers. It is my opinion that the authors have addressed the comments/suggestions in a good manner. I have no further comments or suggestions.

Reviewer 2 Report

Comments and Suggestions for Authors

Accept in present form.